# Effects of Mild Thermal Processing and Storage Conditions on the Quality Attributes and Shelf Life of Truffles (*Terfezia claveryi*)

**DOI:** 10.3390/foods12112212

**Published:** 2023-05-31

**Authors:** Bahareh Daei, Sodeif Azadmard-Damirchi, Afshin Javadi, Mohammadali Torbati

**Affiliations:** 1Department of Food Science and Technology, Mamaghan Branch, Islamic Azad University, Mamaghan 5375113135, Iran; 2Department of Food Science and Technology, Faculty of Agriculture, University of Tabriz, Tabriz 5166616471, Iran; 3Department of Food Hygiene, Faculty of Veterinary, Tabriz Medical Science, Islamic Azad University, Tabriz 5157944533, Iran; 4Department of Food Science and Technology, Faculty of Nutrition and Food Sciences, Tabriz University of Medical Sciences, Tabriz 1561661885, Iran

**Keywords:** brine, mild thermal, preservation, quality, vinegar

## Abstract

This study investigated the effects of two mild thermal processing (MTP) (63 °C, 40 °C, 3 min) methods, in a brine storage medium (7–16% (*w*/*v*) NaCl) and a vinegar solution (5% vinegar, 1% salt, and 0.5% sugar), on some physicochemical properties of truffles (*Terfezia claveryi*). Weight loss, phenolic compounds, firmness, ascorbic acid and microbial loads were evaluated during 160 days of storage. It was demonstrated that a 5% vinegar treatment with 63 °C MTP was effective to reduce the weight loss, microbial spoilage and increased firmness and of truffles during storage. However, phenolic compounds and ascorbic acid content were decreased by heating. Both MTPs inhibited the microbial load, but the 63 °C, 3 min MTP was most effective and resulted in an immediate (3.05–3.2 log CFU/g) reduction in the total aerobic bacteria (TAB) and remained at an acceptable level during storage, while the 40 °C, 3 min MTP reduced (1.12–2 log CFU/g) of the TAB. The results of this study suggest that the 63 °C MTP and immersion in 5% vinegar increased the shelf life of the truffles without perceptible losses in quality attributes.

## 1. Introduction

Truffles are polyphyletic soil fungi that belong to Ascomycota, Basidiomycota, and Zygomycota [1]. Desert truffles are a subclass of mycorrhiza ascomycetes fungi, mainly including species of the genus *Terfezia*, *Tirmania*, *Delastria*, *Tuber* and *Picoa* [2].

Currently, about 30 species of truffles are traded, and due to their unique aroma and potential health benefits, they are one of the most expensive and valuable products in the market [3]. Truffles have been a good source of natural antioxidants such as phenolic and flavonoids compounds, protein, essential amino acids, fatty acids, terpenoids, polysaccharides, minerals, carbohydrates and ergosterols, which have beneficial properties for human health [4]. Some of the potential health benefits of truffles are known, such as anti-tumor, antioxidant, antibacterial, anti-inflammatory, anti-mutagenic and hepatoprotective activities [5].

Truffles are highly perishable and have short postharvest life because of their high metabolic activity and also harvesting at the end of fruit ripening from natural soils, with high pest and microbial load that lead to a short shelf life (7–10 days) [6]. 

Due to their high nutritional value as well as their popularity and high cost, various methods are used to increase the storage time and shelf life. Various preservation techniques have been studied to preserve the physicochemical properties of truffles [7]. These methods cause major changes in the properties of truffles. Truffles used in the food industry require the least amount of heat treatment to preserve their organoleptic properties [8]. Thus, mild heat treatment is required to inactivate enzymes and stabilize the quality with minimal tissue damage and organoleptic properties [9].

Pickling is a common method for preservation of perishable and seasonal leafy vegetables [10]. It has been used for many years to preserve food and extend shelf life. This method involves preserving foods in high acid concentration to maintain texture and flavor during storage [11].

Therefore, the aim of this research aimed to evaluate the effect of different preservation methods including mild thermal processing, vinegar-pickling and immersion in different concentrations of brine on the physicochemical properties and microbial quality of truffles.

## 2. Materials and Methods

### 2.1. Materials

In this experimental research, fresh truffles were collected at maturity from Khoy (Western Azerbaijan Province, Iran) area, during April 2020 and randomly divided into three sets of 3 kg truffle each. Iodized food-grade salt with 99.2% purity and sugar were obtained from a local market (Tabriz, Iran). Commercial grape vinegar (5%) was purchased from Varda Company (Tehran, Iran). All chemicals were analytical grade and purchased from Merck KGaA (Gernsheim, Germany). The culture media of Plate Count Agar was purchased from (PCA, Scharlau Spain) and Potato Dextrose Agar from Sigma-Aldrich Co. (St. Louis, MO, USA).

### 2.2. Methods

#### Treatments and Storage Conditions

Fresh truffles were gently brushed under running tap water to remove soil and debris and then rinsed with sterile water. In order to prepare the brine, sodium chloride and distilled water were used. To prepare samples, cleaned truffles were immersed in different concentrations of brine (7, 10, 13, and 16%) and also in a vinegar solution (5% vinegar, 1% salt, and 0.5% sugar) in jars and sealed. A constant-temperature water bath was used to prepare the MTP treatments. The truffles jars were filled with hot brine or/and hot vinegar solution and then were heated in hot water pot at (63 and 40 °C) 3 min and immediately cooled in an ice-water bucket [12].

### 2.3. Proximate Composition of Fresh Truffles

Proximate composition of truffle samples was determined according to the AOAC methods [13]. In brief, oven-drying was used to determine dry matter of truffles, which were dried at 105 °C for 72 h until constant weight. Crude proteins were measured by the Kjeldahl method by calculating the total nitrogen × 4.38 (nitrogen factor). The Fehling method was used to determine the carbohydrate content. The ash content was also determined by combustion of the samples in a muffle furnace (Nabertherm Muffle Furnace, L 1/12-LT 40/12, Germany) for 4 h at 550 °C. Crude fat was extracted by the Soxhlet method using hexane as an extraction solvent [14].

### 2.4. Determination of the Phenolic Composition of T. claveryi 

Phenolic compounds were analyzed according to the previously published literature [15]. Briefly, the samples were ground in a mixture of acetic acid and methanol (15:85) for 24 h and extraction was then centrifuged. The supernatant was mixed with *n*-hexane and vigorously mixed. Then the bottom solution was separated and 20 μL was injected into the HPLC after filtration. An England CECIL model of high-performance liquid instrument (Cecil CE-4900, Cambridge, England),, equipped with dual pump (Cecil, Cambridge, England)air bubble remover (Cecil) and ultraviolet detector (Cecil, 4201 UV/Vis) with C18 column reversed phase and pore diameter of 5 μm, was used. The phenolic compound peaks were determined by comparing their retention times with that of their reference standards.

### 2.5. Analysis of Ascorbic Acid

Vitamin C content was determined by iodometric titration at the first day (production day) and days 40, 80, 120, and 160 of storage [16] The truffle sample (10 g) was crushed and mixed with 150 mL of distilled water. The mixture was placed in a conical flask (wrapped with aluminum foil) and filtered through a Whatman filter paper No. 4 to obtain a clear extract. After adding 1 mL of starch solution (1%), it was titrated by the iodine solution until the appearance of a blue-black color. The content of vitamin C was calculated by multiplying the volume of iodine solution used for titration by 0.88 mg.

### 2.6. Weight Loss Percentage

The truffles were weighed before pickling and at days 1, 40, 80, 120, and 160 of storage. Weight loss (WL) was determined according to the standard method of AOAC [13] as described below and defined as the percentage of loss of weight of samples with respect to the initial weight:WL (%) = (IW − FW)/IW × 100 (IW): initial weight, (FW): final weights 

### 2.7. Firmness Analysis

A penetration test was performed to determine the tissue firmness of truffles using the previously described method [17]. Truffles were penetrated using a texture analyzer (CF1-250150–STM-1, SANTAM, Tehran. Iran) with a 5 mm cylindrical probe with a speed of 2.0 mm s^−1^. Firmness was defined as the force recorded in a force-time curve obtained from the texture analyzer at a depth of penetration of 5 mm during the compression of the truffle by the cylinder probe. The results of the penetration test were expressed from the time vs. force curves in N mm^−1^. Firmness was determined as the maximum force.

### 2.8. Microbial Analysis

Total aerobic plate counts (APCs) and yeast and molds (Y&M) were determined following the previously described method [18]. The number of total aerobic bacteria was enumerated after homogenization of 5 gr truffle from each pickle jar in 45 mL 1% peptone water with a stomacher at high speed for 2 min and then serially diluted by taking 1.0 mL in 9 mL of peptone water (10^−1^–10^−9^) and pour-plated by using the Plate Count Agar medium (PCA, Scharlau, Spain). The plates were incubated at 35 °C for 48 h and enumerated every 40 days up to 160 days. The results were expressed as log CFU/g. Yeasts and molds (Y&M) were estimated at the same preparation method by using the Potato Dextrose Agar medium (PDA, Sigma-Aldrich). The plates were incubated at 25 °C for 5–7 days.

### 2.9. Statistical Analysis

Statistical analyses were performed using SPSS 18.0 (SPSS Inc., Chicago, IL, USA). Significant differences were analyzed using ANOVA and Duncan’s multiple-range test at a significance level of 0.05. All the experiments were carried out in triplicates. 

## 3. Results and Discussion

### 3.1. Physicochemical Composition of T. claveryi 

*T. claveryi* is a dark brown, oval and potato-shaped fruit body truffle with ivory interior and thin veins with an approximate diameter of 4.5 cm and a length of about 6.5 cm. The collected truffle samples had different weights and volumes. The average weight and volume of *Terfezia claveryi* were 71.5 g and 75.5 cm^3^, respectively (Table 1).

*T. claveryi* contain various nutrients and have been well explored for its chemical profile and nutritional content. The nutritional value of truffles differs from region to region [19]. The average moisture content of the fresh truffle was 74.3 g/100 g, which is in close agreement with the previously reported data for *T. claveryi* [20]. However, some other studies found higher moisture content as high as 79.2–81.6% for truffles of Middle Eastern countries and Arabian truffles [21]. This level of moisture content is also an indication that truffles are highly perishable products. 

Crude fat, crude protein and carbohydrates of samples were 1.4, 11.7 and 14.6%, respectively. The content of crude fat was higher than that reported for *T. claveryi* (0.89 to 1.10%) [21]. A higher amount of crude fat content (3.9%) has been reported for *T. claveryi* of the Iraq region [22] and in another study, it was from 3.5 to 5.5% for same species of truffle harvested from the Northwest of Iran [5]. The protein content of the samples obtained was about 11.7%, which was in agreement with those reported for black truffles (11.9 to 15.95%) [21,23], and which is close to the results reported for *T. claveryi* (13–17%) [5]. However, a lower amount of protein content (3.35%) was also reported previously [24]. The carbohydrate content of the samples obtained was 14.6%, which was lower than the previously reported results (28%) [25]. These differences in nutrition values could be due to the variation of the locations, soil characteristics and climate conditions. 

### 3.2. Influence of Treatments on the Weight Loss 

Weight loss occurs due to the transpiration and respiration of fruits and also, due to the osmosis process as a result of the exchange with the storage solution. Truffles have a moisture content of about 74% and thus they are very susceptible to rapid weight loss accompanied by visible shriveling. After 160 days, a clear increase in weight loss was observed in the control and treated truffles. This was because the membrane permeability increases during the storage period and this led to a decrease in the strength of cell tissue [26]. The results indicated that the rate of weight loss was significantly (*p* < 0.05) slower in truffles processed at 63 °C compared to the control samples (Table 2). Weight loss of the control samples significantly increased (*p* < 0.05), reaching values of 10.2 to 33.84% for different concentration of immersion liquid at the end of storage, while this value was 6.36 to 10.24 in sample treatments at 63 °C. Moreover, the weight loss rate in the 40 °C treated samples was higher than the 63 °C samples. The weight losses of the samples increased in the brine stored samples with increasing the salt concentration. Increasing the concentration of the brine increases the membrane absorption and transfer of the interstitial water to the surrounding fluid due to the high ionic strength, and the reverse osmosis process increases the weight loss of truffles. Additionally, with increasing the salt concentration of the brine and during storage, the tissue strength decreases and the membrane permeability increases, which reduces the weight of the truffles. Relative humidity of the environment and storage temperature are also important because of their influence on the vapor pressure differences between fruit and the surrounding medium.

Among the samples preserved with different methods, the truffles treated at 63 °C and pickled in the 5% vinegar solution had the lowest values of weight loss compared to the 40 °C treated and control non-treated samples during the storage.

### 3.3. Influence of Treatments on Phenolic Compounds 

Phenolics compositions are very important from nutritional and stability points of view. *T. claveryi* is rich in secondary metabolites and antioxidants such as phenolic compounds. Due to the mentioned properties, determination of phenolic compounds is an important factor for the evaluation of *T. claveryi* quality.

The results of the phenolic composition and content of the *Terfezia claveryi* samples are presented in Table 3. Ten phenolic compounds from *T. claveryi* were identified in the fresh truffle samples. Protocatechuic acid was the predominant phenolic acid (26.67 mg/g), followed by gentisic acid, chlorogenic acid, p-hydroxy benzoic acid and other minor phenolic compounds. It has been reported that the analysis of the phenolic contents in the acetone extract of *T. claveryi* has resulted in the phenolic compounds, namely homogentisic acid, protocatechuic acid, gentisic acid, 3,4-dihydroxybenzaldehyde, syringic acid, p-coumaric acid, vanillic acid and some compound with lower amounts [27]. Generally, the obtained results were in agreement with the previously published data [27].

Both methods of thermal processing resulted in a significant reduction (*p* < 0.05) in all phenolic compounds. These compounds decreased significantly with heating and an increase in the salt concentration of the brine. The reduction ranges at 40 °C and 63 °C were 31.3–81% and 34.2–100% during storage, respectively (Table 3). The phenolic content decreased during storage, but reduction was less in the non-heated truffles stored in the vinegar solution. Also, a decrease in the phenolic contents was reported for kidney beans, field peas and chickpeas during the thermal processing [28]. This decrease in the phenolic compounds might be due to the breakdown of phenolics during the heat treatment and leaching of phenolics into the medium solution.

### 3.4. Influence of Treatments on the Texture

Texture is a vital indicator to evaluate the quality of pickled products that is extremely important in overall consumer acceptance. Retention of firmness can be explained by the delay in the breakdown of insoluble protopectin’s into pectic acid and more soluble pectic. The firmness of all the samples decreased as the storage time increased. However, the heated treatment samples remained firmer than the control samples. The firmness of the control samples decreased by 77.8 to 96.6 % at 160 days. As can be seen from Table 4, the firmness of all samples also decreased with increasing of the salt concentration. This is due to the effect of the osmotic dewatering process as a result of the mass transfer in the concentration gradient between the sample and the osmotic solution, which was in line with the previously reported literature [29]. 

Among the samples preserved with the different methods, the truffles processed at 63 °C had a significant tissue score compared to those processed at 40 °C and the control samples. This could be due to the reduced or eliminated activity of the pectin methyl esterase and polygalacturonase enzymes breaking the hydrogen bonds of their three-dimensional structure by heat [30]. After 160 days of storage at 4 °C, the hardness of the vinegared truffles treated with 63 °C was 4.92 N, about 1.57 N higher (*p* < 0.05) than that using the 40 °C heat treatment (3.35 N). This group of treatments achieved the best effect with regard to hardness. Truffles firmness was better maintained during storage if the samples had been pre-treated at 63 °C. Concerning the preservation method, the texture of the samples kept in 5% vinegar and 7% NaCl were in good condition compared with the other samples during storage (Figure 1).

Firmness of the unheated samples decreased almost 96.6% after 160 days of storage and up to 79.1% in the samples treated at 40 °C, while firmness reduction was up to 58% in the samples treated at 63 °C. These results were in agreement with previously published data [31], which reported that mild heating improved the cell structure and firmness with an increase in calcium amounts, leading to the crosslinking of pectin in the carrot samples.

### 3.5. Influence of Treatments on the Ascorbic Acid

The content of ascorbic acid in food products is an important nutritional factor that is usually evaluated and monitored during processing and storage. It was about 4.2 mg/100 g in the fresh truffles at the initial day, which reduced (*p* < 0.05) during the storage for all the preservation treatments (Table 5). However, the reduction rate of ascorbic acid was slower for the samples stored in the refrigerator in comparison to the ones stored at ambient temperature. Ascorbic acid is one of the most unstable vitamins and its content can be reduced during processing and storage due to degradation and Maillard reaction. Oxygen, heat, light, storage time, and storage temperature are factors affecting the ascorbic acid degradation rate [32].

The ascorbic acid content decreased significantly (*p* < 0.05) with the thermal processing and also with increasing the salt concentration in the filling medium. The loss of the ascorbic acid content with increasing the salt concentration could be explained by increasing the osmotic gradient, which causes the transfer of soluble material such as ascorbic acid to the brine [33]. As expected, the reduction in ascorbic acid in the MTP samples during storage is due to the heat stress [34]. This reduction was higher in the samples processed at 63 °C and 40 °C, respectively. Among the samples preserved with different methods, the truffles pickled in the vinegar solution with no heat treatment had the highest ascorbic acid content during the storage.

### 3.6. Microbiological Analysis

The effect of the salt concentration of the brine solutions and the change in aerobic bacteria counts during storage are shown in Table 6. Mild thermal processing is an effective method to eliminate microbial contaminants and extend shelf life. These gentle treatments consist of placing the products at a temperature of 50–90 °C for a period of 1–5 min [35]. The results showed that both thermal treatments of truffles in all groups were able to reduce the microbial flora in the fresh-made samples. However, from these results, 63 °C, 3 min were the most effective to reduce 3.05–3.2 log CFU/g of the total aerobic bacteria (TAB) when compared to the control, while 40 °C, 3 min caused a 1.12–2 log CFU/g reduction of the TAB. The microbial counts were increased in all treatment samples during storage, but the rate of increase in 63 °C MTP was slower than that over the two treated samples. These results indicated that MTP at the 40 °C treatment applied in this study could not have the protective and sufficient effects on the inhibition of microbial spoilage, which led to a further increase in microbial loads during storage.

The effect of MTP and the salt concentrations of the brine on the yeast and molds of the samples during storage is shown in Table 7. Similar results occurred in the yeast and molds reduction. A temperature of 63° C was also able to reduce the yeast and molds load to an acceptable and safe level. 

The results showed that with increasing the salt concentration, the growth of the aerobic bacteria was delayed. After 160 days of storage, the aerobic bacteria count in the thermal processed samples was significantly lower, while the unheated samples showed a sharp increase rate, and the colonies were uncountable at the end of storage with visible spoilage (>8 log CFU/g).

The growth rate of the mesophilic population in the 5% vinegar treatment was lower compared to the brine treatments during storage. It can be due to the antimicrobial role of the high concentrations of vinegar and penetrating the cell wall and denaturing cell plasma proteins. These results are in line with a previous study that reported that acetic acid in vinegar can act as an antimicrobial agent and lead to bacterial cell death [36]. Based on the obtained results, the combined effect of the MTP treatments and immersion in the preservative solutions against microbial contaminations, combined the treatments at 63 °C MTP and immersion in 5% vinegar, appears to be an efficient and sustainable method to ensure microbial safety and to extend the refrigerated shelf life of fresh truffles.

## 4. Conclusions

As our main conclusion, mild thermal processing at 63 °C-3 min can be considered as an efficient and sustainable technique that preserves the quality and organoleptic properties of truffles. Moreover, due to the antimicrobial efficacy of the 5% vinegar and 1% salt solution for the growth inhibition of yeast and mold and enhancement of shelf life up to 160 days, this can be suggested as a safe preservation medium and method for truffles during storage. These observations suggest that mild thermal processing, pickling in 5% vinegar solution can substantially enhance the shelf life of truffles and can be suggested for the marketing of this precious product.

## Figures and Tables

**Figure 1 foods-12-02212-f001:**
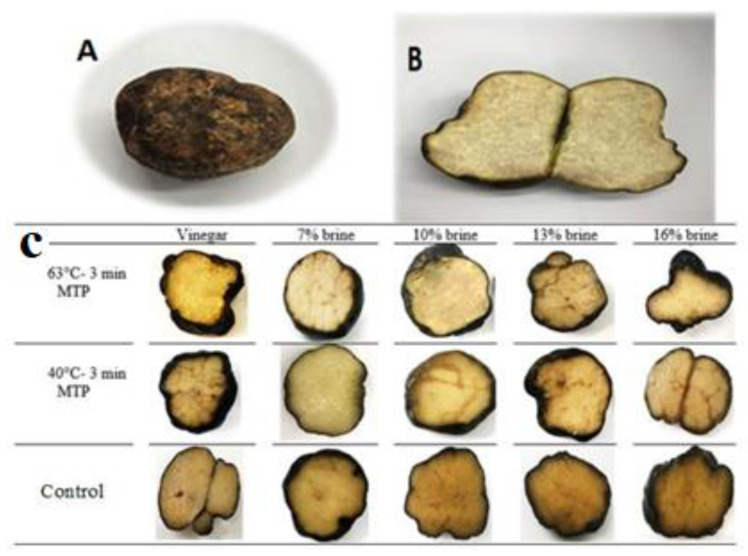
Macro features of *Terfezia claveryi* at first day: (**A**) mature ascocarps, (**B**) white-creamy texture and gleba of mature ascocarp, (**C**): at the end of storage.

**Table 1 foods-12-02212-t001:** Appearance characteristics and proximate composition of truffles (*T. claveryi*).

Factor	Value
Weight (g)	71.5 ± 0.5
Volume (cm^3^)	75.5 ±0.4
Length (cm)	6.5 ± 0.3
Diameter (cm)	4.5 ± 0.2
Moisture (% in DW)	74.3 ± 0.5
Protein (% in DW)	11.7 ± 0.5
Carbohydrates (mg 100 g^−1^ DW)	14.6 ± 0.5
Vitamin C (mg/100 g)	4.2 ± 0.8
Fat (% in DW)	1.4 ± 0.5

**Table 2 foods-12-02212-t002:** Effect of different percentages of brine and type of processing and storage time on the weight loss of truffle samples.

Storage Days (D)						
Preservation Method	Brine (Salt%)	1	40	80	120	160
Control	5% vinegar	0.01 ^Eb^	3.09 ^Dd^	5.92 ^Cd^	8.29 ^Bf^	10.2 ^Af^
	7%	0.01 ^Eb^	3.49 ^Dcd^	7.01 ^Cc^	10.14 ^Be^	13.41 ^Ae^
	10%	0.01 ^Eb^	3.62 ^Dcd^	7.84 ^Cc^	12.27 ^Bd^	15.79 ^Ad^
	13%	0.01 ^Eb^	4.29 ^Dbc^	9.20 ^Cb^	14.44 ^Bc^	18.65 ^Ac^
	16%	0.01 ^Eb^	7.60 ^Da^	15.85 ^Ca^	24.49 ^Ba^	33.84 ^Aa^
MTP 40 °C-3 min	5% vinegar	0.008 ^Ec^	2.19 ^Dd^	4.04 ^Cef^	6.39 ^Bg^	8.19 ^Ag^
	7%	0.008 ^Ec^	2.53 ^Dcd^	4.72 ^Ce^	7.29 ^Bf^	9.64 ^Af^
	10%	0.009 ^Ec^	2.60 ^Dc^	5.15 ^Cde^	8.14 ^Bf^	10.41 ^Af^
	13%	1.01 ^Ea^	3.59 ^Dc^	5.64 ^Cd^	12.21 ^Bd^	15.5 ^Ad^
	16%	1.09 ^Ea^	4.84 ^Db^	10.34 ^Cb^	19.84 ^Bb^	26.77 ^Ab^
MTP 63 °C-3 min	5% vinegar	0.003 ^Ec^	1.46 ^Dd^	3.57 ^Cf^	5.19 ^Bg^	6.37 ^Ah^
	7%	0.005 ^Ec^	1.62 ^Dd^	3.68 ^Cf^	5.44 ^Bg^	6.74 ^Ah^
	10%	0.005 ^Ec^	2.09 ^Dd^	3.98 ^Cf^	6.27 ^Bg^	7.14 ^Ag^
	13%	0.008 ^Ec^	2.53 ^Dcd^	4.72 ^Ce^	7.29 ^Bf^	9.01 ^Af^
	16%	0.009 ^Ec^	4.69 ^Db^	6.7 ^Ccd^	8.14 ^Bf^	10.25 ^Af^

Different lowercase letters in each column and different uppercase letters in each row indicate a significant difference at the 5% level.

**Table 3 foods-12-02212-t003:** Effect of different percentages of brine and type of processing and storage time on the phenolic content (μg/g dry weight of *T. claveryi*).

Time	Treatment		Catechin	p-Coumaric Acid	Ferulic Acid	Chlorogenic Acid	Syringic Acid	p-Hydroxy Benzoic Acid	Rutin	Proto Catechuic Acid	Eugenol	Gentisic Acid
Day 1			8.29 ^A^**	14.48 ^A^	12.15 ^A^	24.67 ^A^	2.47 ^A^	25.49 ^A^	5.58 ^A^	26.67 ^A^	8.45 ^A^	26.40 ^A^
		A *	5.70 ^BC^	9.10 ^CD^	8.05 ^C^	19.25 ^CD^	1.75 ^D^	18.54 ^D^	3.18 ^CD^	19.87 ^C^	6.11 ^C^	16.70 ^DE^
	5% vinegar	B	5.45 ^C^	8.29 ^D^	7.10 ^D^	17.48 ^E^	1.00 ^I^	16.20 ^G^	2.88 ^D^	16.45 ^F^	5.03 ^E^	14.55 ^G^
		C	6.15 ^B^	10.18 ^B^	9.40 ^B^	20.85 ^B^	2.12 ^B^	20.05 ^B^	4.10 ^B^	22.20 ^B^	6.80 ^B^	20.35 ^B^
		A	4.64 ^CD^	6.83 ^F^	6.50 ^DE^	16.20 ^FG^	1.64 ^DE^	17.64 ^F^	2.30 ^E^	18.30 ^D^	5.85 ^D^	14.78 ^G^
	7% Brine	B	3.20 ^E^	5.35 ^I^	5.48 ^EF^	13.10 ^H^	0.96 ^I^	14.18 ^I^	1.96 ^EF^	15.64 ^G^	4.94 ^E^	13.40 ^H^
		C	5.65 ^BC^	8.45 ^D^	7.87 ^CD^	20.18 ^BC^	2.05 ^C^	19.40 ^C^	3.70 ^BC^	21.80 ^B^	6.54 ^C^	18.65 ^CD^
		A	3.75 ^E^	6.10 ^FG^	6.15 ^E^	15.35 ^G^	1.20 ^H^	15.40 ^H^	2.18 ^E^	17.65 ^E^	5.67 ^DE^	14.25 ^GH^
Day 160	10% Brine	B	2.48 ^F^	5.18 ^GH^	5.12 ^F^	12.49 ^I^	0.58 ^J^	12.55 ^K^	1.40 ^F^	14.80 ^GH^	4.48 ^FG^	12.18 ^I^
		C	5.25 ^BC^	8.10 ^DE^	7.68 ^CD^	19.46 ^C^	1.87 ^CD^	19.18 ^CD^	3.45 ^C^	21.18 ^BC^	6.30 ^C^	18.45 ^CD^
		A	3.64 ^E^	5.84 ^G^	5.45 ^EF^	14.97 ^GH^	1.10 ^H^	13.45 ^J^	2.06 ^E^	16.30 ^F^	5.10 ^E^	12.40 ^I^
	13% Brine	B	1.83 ^G^	4.96 ^I^	4.55 ^FG^	11.87 ^J^	0.35 ^J^	11.80 ^L^	1.26 ^F^	11.28 ^I^	4.12 ^G^	12.28 ^I^
		C	4.67 ^CD^	8.03 ^DE^	7.40 ^D^	19.16 ^CD^	1.63 ^DE^	18.96 ^CD^	3.28 ^CD^	20.15 ^C^	6.18 ^C^	17.36 ^D^
		A	2.15 ^F^	5.15 ^GH^	4.98 ^FG^	14.40 ^GH^	0.45 ^J^	12.15 ^KL^	1.43 ^F^	14.84 ^GH^	4.65 ^F^	12.19 ^I^
	16% Brine	B	1.10 ^H^	4.80 ^I^	4.10 ^G^	11.25 ^J^	ND	10.50 ^M^	1.10 ^FG^	10.15 ^I^	3.30 ^H^	11.85 ^J^
		C	3.28 ^E^	7.30 ^E^	6.80 ^DE^	18.38 ^D^	1.50 ^F^	18.80 ^CD^	2.65 ^DE^	19.43 ^CD^	5.80 ^D^	16.80 ^DE^

*: (A) MTP at 40 °C-3 min, (B) MTP at 63 °C-3 min, (C) Unheated control samples. **: Different capital letters in column indicate a significant difference in the probability level of 5%.

**Table 4 foods-12-02212-t004:** Effect of different percentages of brine and type of processing and storage time on the texture of truffle samples.

Storage Days (D)						
Preservation Method	Brine (Salt%)	1	40	80	120	160
Control	5% vinegar	6.78 ^Aa^	4.78 ^Bbc^	3.69 ^Bbc^	2.48 ^Cd^	1.5 ^Dd^
	7%	6.78 ^Aa^	4.0 ^Bc^	2.88 ^Ccd^	2.29 ^Dd^	1.19 ^Ed^
	10%	6.78 ^Aa^	3.79 ^Bc^	2.63 ^Ccd^	1.83 ^Dde^	0.96 ^Ed^
	13%	6.78 ^Aa^	3.68 ^Bc^	2.23 ^Cd^	1.19 ^De^	0.91 ^Dd^
	16%	6.78 ^Aa^	3.41 ^Bd^	1.9 ^Cd^	0.93 ^De^	0.23 ^De^
MTP 40 °C-3 min	5% vinegar	6.78 ^Aa^	5.81 ^Bab^	4.29 ^Cb^	3.69 ^Dbc^	3.35 ^Dbc^
	7%	6.81 ^Aa^	5.38 ^Bb^	4.08 ^Cb^	3.56 ^Dbc^	3.18 ^Dbc^
	10%	6.79 ^A^	4.63 ^Bb^	3.78 ^Cbc^	2.43 ^Dd^	2.0 ^Dd^
	13%	6.73 ^Aab^	4.13 ^Bcd^	2.66 ^Ccd^	2.23 ^CDd^	1.83 ^Dd^
	16%	6.76 ^Aa^	4.03 ^Bc^	2.19 ^Cd^	1.78 ^CDde^	1.41 ^Dd^
MTP 63 °C-3 min	5% vinegar	6.76 ^Aa^	6.21 ^ABa^	5.8 ^Ba^	5.33 ^BCa^	4.92 ^Ca^
	7%	6.73 ^Aab^	5.78 ^Bab^	5.38 ^BCab^	4.96 ^Cab^	4.74 ^Ca^
	10%	6.76 ^Aa^	5.7 ^Bab^	5.08 ^BCab^	4.31 ^Cb^	3.71 ^Db^
	13%	6.69 ^Ab^	5.11 ^Bb^	4.39 ^Cb^	3.82 ^Dbc^	3.22 ^Dbc^
	16%	6.73 ^Aab^	4.1 ^Bc^	3.48 ^Cc^	3.09 ^Cc^	2.53 ^Dc^

Different lowercase letters in each column and different uppercase letters in each row indicate a significant difference at the 5% level.

**Table 5 foods-12-02212-t005:** Effect of different percentages of brine and type of processing and storage time on the content of ascorbic acid of truffle samples.

Storage Time (Day)						
Preservation Method	Brine (Salt%)	1	40	80	120	160
Control	5% vinegar	4.43 ^Aa^	3.48 ^Ba^	2.83 ^Ca^	2.41 ^CDa^	1.66 ^Da^
	7%	4.41 ^Aa^	3.5 ^Ba^	2.68 ^Ca^	2.24 ^CDa^	1.51 ^Da^
	10%	4.38 ^Aa^	3.33 ^Ba^	2.38 ^Cab^	2.03 ^Ca^	1.18 ^Da^
	13%	4.41 ^Aa^	3.08 ^Bab^	2.13 ^Cb^	1.88 ^CDab^	1.1 ^Dab^
	16%	4.38 ^Aa^	3.03 ^Bab^	2.01 ^Cb^	1.78 ^CDab^	1.03 ^Da^
MTP 40 °C-3 min	5% vinegar	4.48 ^Aa^	3.28 ^Bab^	2.44 ^Cab^	2.10 ^Da^	1.38 ^Ea^
	7%	4.48 ^Aa^	3.13 ^Bab^	2.28 ^Cab^	1.98 ^Dab^	1.23 ^Ea^
	10%	4.48 ^Aab^	3.10 ^Bab^	2.13 ^Cb^	1.76 ^Dab^	1.04 ^Ea^
	13%	4.48 ^Aa^	3.08 ^Bab^	2.00 ^Cb^	1.61 ^Db^	0.92 ^Eb^
	16%	4.48 ^Aa^	2.82 ^Bb^	1.92 ^Cbc^	1.48 ^CDb^	0.81 ^Db^
MTP 63 °C-3 min	5% vinegar	3.78 ^Ab^	3.13 ^Bab^	2.30 ^Cab^	1.94 ^CDab^	1.33 ^Da^
	7%	3.38 ^Ac^	3.03 ^Aab^	2.16 ^Bb^	1.72 ^BCab^	0.98 ^Cb^
	10%	3.34 ^Ac^	2.78 ^Bb^	2.02 ^Cb^	1.33 ^Db^	0.92 ^Db^
	13%	3.36 ^Ac^	2.43 ^Bb^	1.81 ^BCbc^	1.13 ^Cb^	0.76 ^Db^
	16%	3.40 ^Ac^	2.18 ^Bc^	1.68 ^BCc^	0.96 ^Cb^	0.63 ^Cb^

Different lowercase letters in each column and different uppercase letters in each row indicate a significant difference at the 5% level.

**Table 6 foods-12-02212-t006:** Effect of different percentages of brine and type of processing and storage time on the total count of truffles of truffle samples.

Storage Days (D)						
Preservation Method Temperature	Brine (Salt%)	1	40	80	120	160
Control	5% vinegar	4.08 ^Da^	4.16 ^Db^	4.78 ^Cbc^	5.28 ^Bc^	7.53 ^Ab^
	16%	4.13 ^Da^	4.22 ^Db^	5.29 ^Cab^	5.98 ^Bb^	8.83 ^Ab^
	13%	4.11 ^Ea^	4.26 ^Dab^	5.43 ^Cab^	6.18 ^Bb^	8.65 ^Ab^
	10%	4.16 ^Ea^	4.78 ^Da^	5.83 ^Ca^	6.78 ^Bab^	8.38 ^Ab^
	7%	4.13 ^Ea^	4.92 ^Da^	6.13 ^Ca^	7.33 ^Ba^	8.23 ^Aa^
MTP 40 °C	5% vinegar	2.08 ^Ca^	2.53 ^BCb^	3.05 ^Bbc^	3.86 ^ABc^	4.15 ^Ad^
	16%	2.96 ^Ca^	3.92 ^Ba^	4.46 ^Bab^	4.84 ^Ab^	5.78 ^Ac^
	13%	2.88 ^Ca^	3.78 ^BCa^	4.33 ^Bab^	4.66 ^ABbc^	5.78 ^Ac^
	10%	2.76 ^Ca^	3.63 ^Bab^	4.18 ^BCc^	4.58 ^ABbc^	5.55 ^Ac^
	7%	2.63 ^Ba^	3.56 ^Bab^	3.96 ^Bc^	4.24 ^ABbc^	5.51 ^Ac^
MTP 63 °C	5% vinegar	0.88 ^Cb^	1.52 ^Bb^	2.23 ^Bb^	2.86 ^ABd^	3.03 ^Ad^
	16%	1.26 ^Cb^	2.73 ^Ba^	3.50 ^ABab^	3.73 ^Abc^	3.83 ^Acd^
	13%	1.24 ^Cb^	2.46 ^Bab^	3.26 ^ABab^	3.68 ^ABc^	3.78 ^Acd^
	10%	1.16 ^Cb^	2.33 ^Bb^	3.15 ^ABb^	3.58 ^Ac^	3.73 ^Ad^
	7%	1.13 ^Cb^	2.18 ^Bb^	3.03 ^ABbc^	3.53 ^Ac^	3.70 ^Ad^

Different lowercase letters in each column and different uppercase letters in each row indicate a significant difference at the 5% level.

**Table 7 foods-12-02212-t007:** Effect of different percentages of brine and type of processing and storage time on the yeast and mold count of truffles of truffle samples.

Storage Days (D)							
Preservation Method Temperature	Brine (Salt%)		1	40	80	120	160
Control		5% vinegar	3.10 ^Da^	3.66 ^Db^	3.95 ^Cbc^	4.00 ^Bc^	4.15 ^Ab^
		16%	3.13 ^Ea^	3.98 ^Da^	4.84 ^Ca^	4.73 ^Ba^	4.85 ^Aa^
		13%	3.16 ^Ea0^	3.98 ^Da^	4.63 ^Ca^	4.68 ^Bab^	4.78 ^Ab^
		10%	3.11 ^Ea^	3.96 ^Dab^	4.45 ^Cab^	4.56 ^Bb^	4.68 ^Ab^
		7%	3.13 ^Da^	3.92 ^Db^	4.15 ^Cab^	4.38 ^Bb^	4.63 ^Ab^
MTP 40 °C		5% vinegar	0.63 ^Ca^	0.70 ^BCb^	0.76 ^Bbc^	0.86 ^ABc^	0.92 ^Ad^
		16%	0.95 ^Ca^	0.99 ^Ba^	1.16 ^Bab^	1.24 ^Ab^	1.28 ^Ac^
		13%	0.88 ^Ca^	0.95 ^BCa^	0.98 ^Bab^	1.06 ^ABbc^	1.12 ^Ac^
		10%	0.76 ^Ca^	0.83 ^Bab^	0.86 ^BCc^	0.98 ^ABbc^	1.06 ^Ac^
		7%	0.68 ^Ba^	0.76 ^Bab^	0.85 ^Bc^	0.91 ^ABbc^	0.95 ^Ac^
MTP 63 °C		5% vinegar	0.45 ^Cb^	0.50 ^Bb^	0.56 ^Bb^	0.65 ^ABd^	0.68 ^Ad^
		16%	0.65 ^Cb^	0.73 ^Ba^	0.77 ^ABab^	0.82 ^Abc^	0.88 ^Acd^
		13%	0.63 ^Cb^	0.66 ^Bab^	0.74 ^ABab^	0.78 ^ABc^	0.85 ^Acd^
		10%	0.56 ^Cb^	0.60 ^Bb^	0.65 ^ABb^	0.68 ^Ac^	0.78 ^Ad^
		7%	0.53 ^Cb^	0.58 ^Bb^	0.63 ^ABbc^	0.67 ^Ac^	0.75 ^Ad^

Different lowercase letters in each column and different uppercase letters in each row indicate a significant difference at the 5% level.

## Data Availability

Data are contained within the article.

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
