# Peer review of "Effects of Mild Thermal Processing and Storage Conditions on the Quality Attributes and Shelf Life of Truffles (Terfezia claveryi)"

_foods, 2023, doi:10.3390/foods12112212_

Round 1

Reviewer 1 Report

This manuscript investigated the effect of different preservation methods including mild thermal heating, vinegar-pickled and immersion in the different concentration of brine on the physicochemical and microbial properties of truffles. The results suggested that hurdle of mild thermal processing, pickling in 5% vinegar solution could substantially enhance the shelf life of truffles. This manuscript is innovative. However, there are some details in the manuscript that need to be improved. So, I recommend major revisions.

Abstract

1. Some “oC” are not in the right format, please check them carefully.

2. Line 11, “63 ºC MTP and immersion iv 5% vinegar”, what’s meaning of “iv”?

Section 3.1

3. “However, some other studies found higher moisture content as high as 79.2 % for truffles of Middle Eastern countries and Arabian truffles” and “The content of crude fat was higher than that reported for T. claveryi (0.89 to 1.10%)”. Please add references.

4. “The results indicated that rate of weight loss was significantly (p < 0.05) slower in the MTP samples (Table 2).” However, Weight loss rate of MPT (63 oC-3min) significantly higher than the Control after 40, 60, 80, 120 and 160 storage days when the salt concentration of brines was 16%. Similarly, the latter “Moreover, that rate of weight loss was significantly (p < 0.05) slower in the (63° C, 3 min) MTP samples compared with (40° C, 3 min) treated samples.” statement is not accurate.

5. “Weight loss of control samples significantly increased (p<0.05), reaching values of 14.19 and 39.47 %, after 160 storage days”, where did that come from?

6. Table 2, 3 and 4, “MTP 63 oC- min” should revised toMTP 63 oC-3min”. And 160 days is not at the same height as the other days.

7. In Section 3.4, “63 °C MTP samples had a significant tissue score compared 40 °C MTP and control samples, respectively.” However, they were had not a significant different through Table 4.

8. Why are there two Tables 3 in Section 3.3 and 3.5?

9. In the conclusion, “As main conclusion, mild thermal processing at (63 °C, 3 min) minutes…”. The “minutes” should be deleted.

Author Response

Dear Editor and Dear Reviewers:

Authors would like to thank you very much for your precious time and valuable comments and suggestion on the manuscript (foods-2037842). The manuscript was extensively revised according to your valuable opinions and we hope we have fulfilled your requests. All changes were made in red to be easily followed. Response to the comments come as below:

Comments to the Author

Comment: 1. Some “ºC” are not in the right format, please check them carefully.

Response: Thank you very much for this valuable comment. The grammatical mistakes were corrected as advised.

  1. Line 11, “63 ºC MTP and immersion iv 5% vinegar”, what’s meaning of “iv”?

Response: Thank you very much for this valuable comment. The grammatical mistakes were corrected as advised

  1. “However, some other studies found higher moisture content as high as 79.2 % for truffles of Middle Eastern countries and Arabian truffles” and “The content of crude fat was higher than that reported for T. claveryi (0.89 to 1.10%)”. Please add references.

Response: Thank you very much for this valuable comment. This was clarified and referenced in the text page 8 line 155.

  1. “Weight loss of control samples significantly increased (p < 0.05), reaching values of 14.19 and 39.47 %, after 160 storage days”, where did that come from?

Authors thank the reviewer very much for this valuable comment. The grammatical mistakes were corrected as advised. According the table 2 weight loss of control samples significantly increased (p<0.05), reaching values of 10.2 and 33.84 %, after 160 storage days while this value was 6.36-10.24 at 63 °C- 3 min samples.

  1. “The results indicated that rate of weight loss was significantly (p < 0.05) slower in the MTP samples (Table 2).” However, Weight loss rate of MPT (63 oC-3min) significantly higher than the Control after 40, 60, 80, 120 and 160 storage days when the salt concentration of brines was 16%. Similarly, the latter “Moreover, that rate of weight loss was significantly (p < 0.05) slower in the (63 °C, 3 min) MTP samples compared with (40° C, 3 min) treated samples.” statement is not accurate.

Response: Authors thank you very much for this accurate comment. Actually, there was a misplacing in the data presented in the Table 2 which was corrected now. This was also explained in the text as well.

  1. Table 2, 3 and 4, “MTP 63 °C - min” should revised to “MTP 63 °C -3min”. And 160 days is not at the same height as the other days.

Authors thank the reviewer very much for this valuable comment. The grammatical mistakes were corrected as advised.

  1. In Section 3.4, “63 °C MTP samples had a significant tissue score compared 40 °C MTP and control samples, respectively.” However, they were had not a significant different through Table 4.

Response: Thank you very much for this valuable comment. Authors rechecked the results. There were significant differences among the samples for samples in day 160 as it is clear from results and data presented in Table 4

  1. Why are there two Tables 3 in Section 3.3 and 3.5?

Response: Authors thank you very much for your comment. This was corrected and revised as your comment.

  1. In the conclusion, “As main conclusion, mild thermal processing at (63 °C, 3 min) minutes…”. The “minutes” should be deleted.

Response: Thank you very much for this valuable comment. The grammatical mistakes were corrected as advised

Reviewer 2 Report

 The work and design of experiment is good but the manuscript needs extensive improvement.

The corrections have been marked in the manuscript. kindly go through whole manuscript thoroughly for language. Table need attention. Required correction have been given in the attached file. Material and methods must contain the basic of analysis such as size, replication, kind of test used for analysis. P value column and row wise can also be included to improve the clarity in level of significance. All tables have one or more corrections in the superscript. Result has been discussed well. Conclusion clear.

Reference: check for journal guidelines

Author Response

Dear Editor and Dear Reviewers:

 Authors would like to thank you very much for your precious time and valuable comments and suggestion on the manuscript (foods-2037842). The manuscript was extensively revised according to your valuable opinions and we hope we have fulfilled your requests. All changes were made in red to be easily followed and the revised manuscript file comes as attachment. 

Thank you very much again.

With best regards,

Sodeif Azadmard-Damirchi

Round 2

Reviewer 1 Report

Through the modification, the article has been improved. This article is recommended for acceptance.

Author Response

Dear reviewer,

Authors wold like to thank you very much for your valuable helps and comments to improve the manuscript to be at acceptable level.

With best regards,

S. Azadmard-Damirchi

Reviewer 2 Report

The corrections have been incorporated and the manuscript has been improved

Author Response

Dear reviewer,

Authors wold like to thank you very much for your valuable time, helps, suggestions and comments to improve the manuscript. The purity of the salt and the full name of the company were corrected and revised as advised. The revised manuscript comes as an attached file.

With best regards,

S. Azadmard-Damirchi
